# The Utility of Combined Target and Systematic Prostate Biopsies in the Diagnosis of Clinically Significant Prostate Cancer Using Prostate Imaging Reporting and Data System Version 2 Based on Biparametric Magnetic Resonance Imaging

**Daiki Kato** [1], **Kaori Ozawa** [2], **Shinichi Takeuchi** [2], **Makoto Kawase** [1], **Kota Kawase** [1], **Chie Nakai** [1], **Manabu Takai** [1], **Koji Iinuma** [1], **Keita Nakane** [1], **Hiroki Kato** [3], **Masayuki Matsuo** [3], **Natsuko Suzui** [3], **Tatsuhiko Miyazaki** [3] **and Takuya Koie** [1,*]

1   Department of Urology, Gifu University Graduate School of Medicine, Gifu 5011194, Japan; andreas7@gifu-u.ac.jp (D.K.); buki21211128@gmail.com (M.K.); stnf55@gmail.com (K.K.); chie.johha@gmail.com (C.N.); takai_mb@gifu-u.ac.jp (M.T.); kiinuma@gifu-u.ac.jp (K.I.); keitaco@gifu-u.ac.jp (K.N.)

2   Department of Urology, Ogaki Municipal Hospital, Ogaki 5038502, Japan; k.ohzawa.lily@gmail.com (K.O.); gallxy7@gmail.com (S.T.)

3   Department of Radiology, Gifu University Graduate School of Medicine, Gifu 5011194, Japan; hkato@gifu-u.ac.jp (H.K.); matsuo_m@gifu-u.ac.jp (M.M.); nsuzui7@gifu-u.ac.jp (N.S.); tats_m@gifu-u.ac.jp (T.M.)

*   Correspondence: goodwin@gifu-u.ac.jp

**Abstract:** This study aimed to determine the predictive value of the Prostate Imaging Reporting and Data System version 2 (PI-RADS v2) based on biparametric magnetic resonance imaging (bpMRI) with combined target biopsy (TBx) and systematic biopsy (SBx) in patients with suspicion of having clinically significant prostate cancer (csPCa). In this retrospective study, we reviewed the clinical and pathological records of 184 consecutive patients who underwent bpMRI before prostate biopsy. We focused on patients with PI-RADS v2 scores ≥ 3. MRI was performed using a 3-Tesla clinical scanner with a 32-channel phased-array receiver coil. PI-RADS v2 was used to describe bpMRI findings based on T2-weighted imaging and diffusion-weighted imaging scores. The primary endpoint was the diagnostic accuracy rate of PI-RADS v2 based on bpMRI for patients with prostate cancer (PCa) who underwent combined TBx and SBx. A total of 104 patients were enrolled in this study. Combined TBx and SBx was significantly superior to either method alone for PCa detection in patients with suspicious lesions according to PI-RADS v2. TBx and SBx detected concordant csPCa in only 24.1% of the patients. In addition, the rate of increase in the Gleason score was similar between SBx (41.5%) and TBx (34.1%). The diagnostic accuracy of bpMRI is comparable to that of standard multiparametric MRI for the detection of csPCa. Moreover, combined TBx and SBx may be optimal for the accurate determination of csPCa diagnosis, the International Society of Urological Pathology grade, and risk classification.

**Keywords:** prostate imaging reporting and data system; prostate biopsy; prostate cancer; positive predictive value

## 1. Introduction

Prostate cancer (PCa) is the fourth most common malignancy in men [1]. The prostate-specific antigen (PSA) test is widely used as the conventional PCa screening test. According to the European Association of Urology guidelines, 10–12 core systematic transrectal ultrasound (TRUS)–guided biopsies should be performed for patients with elevated PSA levels or abnormal digital rectal examination [2]. However, this diagnostic strategy is disadvantageous because it is based on random sampling and is largely operator dependent [3]. In addition, earlier studies reported that TRUS-guided biopsy missed a substantial

proportion (up to 20%) of clinically significant PCa (csPCa) because of sampling errors [4], and the 30-day complication rates were relatively high [5]. Therefore, any non-invasive examination that can reduce the number of unnecessary biopsies with negative results is worth considering [6].

Recently, multiparametric magnetic resonance imaging (mpMRI) was introduced to improve tumor detection and localization [7]. It is defined as a combination of anatomical imaging techniques consisting of at least two functional modalities including T2-weighted imaging (T2WI), diffusion-weighted imaging (DWI), and dynamic contrast-enhanced MRI (DCE-MRI) [8]. In 2012, the European Society of Urogenital Radiology proposed the Prostate Imaging Reporting and Data System (PI-RADS) to assess the risk of PCa in lesions detected by mpMRI. Subsequently, PI-RADS version 2 (PI-RADS v2) was established in 2015, and it simplified the rules for reporting modified imaging sequences and defined csPCa using T2WI and DWI [9]. However, the role of DCE-MRI in the diagnosis of PCa is controversial. Several studies reported that DCE-MRI played little or no part in the detection of PCa [10,11]. Therefore, recent studies proposed biparametric MRI (bpMRI) without DCE-MRI for PI-RADS v2 [10–12]. Kuhl et al. reported that the diagnostic accuracies of bpMRI and mpMRI were equal when 542 patients with PSA levels $\geq$ 3 after negative pre-biopsy results were assessed [12].

Target biopsy (TBx) of suspicious lesions on mpMRI has produced more favorable csPCa detection rates than TRUS-guided systematic biopsy (SBx) [13]. However, it is possible to miss csPCa using TBx alone [14]. Therefore, current guidelines still recommend combined TBx and SBx for patients with suspicious PCa lesions [15].

The aim of this study was to evaluate the predictive value of PI-RADS v2 based on bpMRI for patients with csPCa who underwent combined TBx and SBx.

## 2. Materials and Methods

### 2.1. Patients

In this retrospective study, we reviewed the clinical and pathological records of 184 consecutive patients who underwent bpMRI before prostate biopsy between August 2016 and June 2019 at Gifu University. We focused on patients with PI-RADS v2 scores $\geq$ 3 who underwent prostate biopsy. Patients with lymph node involvement, distant metastases, or clinical stage T4 tumors according to the 2010 American Joint Committee on Cancer Staging Manual were excluded from the study [16]. Patients who had prostate abscess, patients who received finasteride or dutasteride before prostate biopsy, patients who had previously undergone transurethral prostate resection, and patients who underwent active surveillance (AS) were also excluded.

The study protocol was approved by the Institutional Review Board of Gifu University (number: 30-031).

### 2.2. bpMRI Protocol

All the patients enrolled in this study underwent bpMRI before prostate biopsy. MRI was performed using a 3-Tesla (T) clinical scanner with a 32-channel phased-array receiver coil (Ingenia CX; Philips Healthcare, Best, Veenpluis, The Netherlands). PI-RADS v2 was used to describe the MRI findings based on the T2WI and DWI scores. At least two experienced radiologists (with more than 5 years of experience in PCa diagnosis) were assigned to review the bpMRI. In this study, the highest overall PI-RADS v2 score based on bpMRI was used in the analysis of each patient, irrespective of the prostate zone.

### 2.3. Prostate Biopsy Protocol

The prostate biopsies were performed by six surgeons. TRUS-guided transrectal or transperineal systematic biopsy was performed using an 18-G automatic biopsy gun (PRIMECUT®, Boston Scientific, Marlborough, MA, USA) under spinal anesthesia. In all the patients, 12 cores (eight in the peripheral zone (PZ) and four in the transitional zone (TZ)) were biopsied for SBx. Each TBx was performed using MRI-TRUS fusion biopsy

(HI VISION Ascendus Sonography system, Hitachi Medical Corporation, Tokyo, Japan). Suspicious lesions on bpMRI (PI-RADS v2 scores $\geq$ 3) were generally targeted, with two to four cores depending on lesion size. The prostate biopsy specimens were evaluated by a single pathologist at our institution.

### 2.4. csPCa

csPCa was defined by the International Society of Urological Pathology (ISUP) grade $\geq$ 2 and/or a maximum cancer core length $\geq$ 4 mm within at least one specimen, which was obtained after MRI-TRUS fusion biopsies, whereas clinically insignificant cancer was defined as group 1 according to the 2014 ISUP guidelines [6,17].

### 2.5. Statistical Analysis

The primary endpoint was the diagnostic accuracy rate of PI-RADS v2 based on bpMRI for patients with PCa, which includes csPCa and clinically insignificant PCa, who underwent combined TBx and SBx. The secondary endpoints were the csPCa detection rates of TBx and SBx, the positive predictive values (PPVs) of the different PI-RADS groups using combined TBx and SBx, and the diagnostic accuracy of PI-RADS v2 compared to that of histological findings. Continuous variables were compared using Student's *t*-test, and categorical variables were compared using Fisher's exact test or the McNemar test. All *p* values were two-sided, and *p* values <0.05 were considered statistically significant.

## 3. Results

### 3.1. Patient Characteristics

A total of 104 patients were enrolled in this study. We excluded patients with lymph node involvement (*n* = 1), patients who received dutasteride before prostate biopsy (*n* = 2), and patients who underwent AS (*n* = 2). The demographic data of the enrolled patients are shown in Table 1.

**Table 1.** Patient characteristics.

| | |
|---|---|
| Age (year, median, interquartile range) | 71 (67–75) |
| Body mass index (kg/m$^2$, median, interquartile range) | 23.2 (21.5–25.3) |
| Prostate-specific antigen (ng/mL, median, interquartile range) | 8.32 (5.38–13.86) |
| Prostate volume (mL, median, interquartile range) | 30.7 (23.0–44.8) |
| Prostate-specific antigen density (ng/mL/cm$^3$, median, interquartile range) | 0.26 (0.18–0.44) |
| Prostate Imaging Reporting and Data System version 2 (number, %) | |
| 3 | 22 (21.1) |
| 4 | 55 (52.9) |
| 5 | 27 (26.0) |

### 3.2. PCa Detection

A total of 78 patients (75.0%) were diagnosed with PCa using prostate biopsy. According to the PCa detection rate by the differences of prostate biopsy, combined TBx and SBx was significantly superior to TBx (69.2%) or SBx (63.5%) alone in PCa detection in patients with suspicious lesions according to PI-RADS v2 (*p* = 0.001, *p* < 0.001, respectively).

According to the PCa detection, the association between PI-RADS v2 and ISUP grade is shown in Figure 1. PI-RADS v2 scores were found to significantly correlate with ISUP grade (*p* = 0.004; Figure 1). PPVs of TBx were similar in the PZ and TZ of patients with PI-RADS v2 scores $\geq$ 3 (60.9% and 52.1%; *p* = 0.448), $\geq$4 (64.3% and 60.6%; *p* = 0.821), and 5 (77.8% and 75.0%; *p* > 0.999).

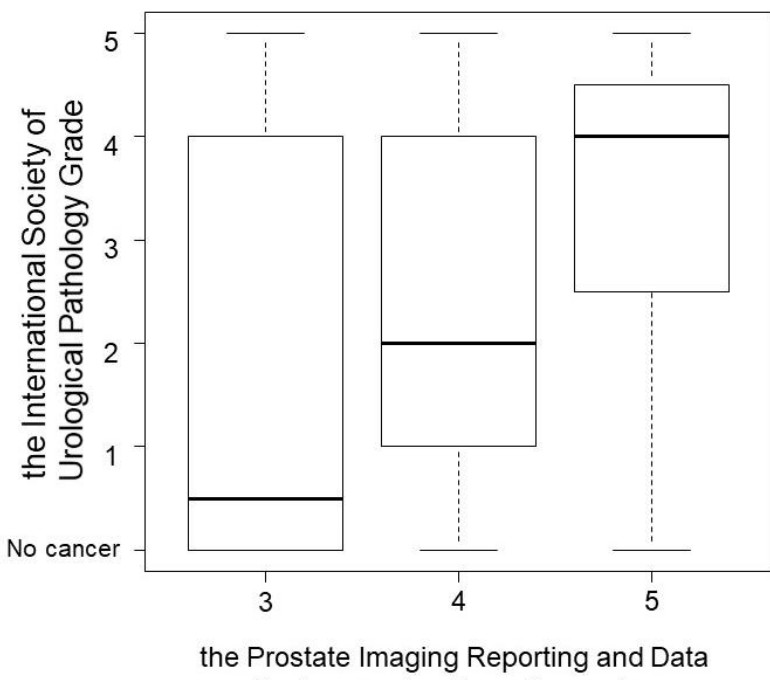

**Figure 1.** The association between Prostate Imaging Reporting and Data System version 2 (PI-RADS v2) and International Society of Urological Pathology (ISUP) grade according to the prostate cancer detection. PI-RADS v2 scores were found to significantly correlate with ISUP grade ($p = 0.004$).

As shown in Table 2, TBx and SBx detected concordant csPCa in only 24.1% of patients. In addition, the rate of increase in the Gleason score (GS) was similar between SBx (41.5%) and TBx (34.1%).

**Table 2.** ISUP grade and prostate cancer location concordance outcomes according to target or systemic biopsies in patients with clinically significant prostate cancer.

|  | Number, % |
| --- | --- |
| Concordance | 13 (24.1) |
| Inconsistency | 41 (75.9) |
| Concordance ISUP grade between target and systemic biopsy | 10 (24.4) |
| Upgrading ISUP grade on target biopsy | 14 (34.1) |
| Upgrading ISUP grade on systemic biopsy | 17 (41.5) |

Table 3 shows the complications according to the type of prostate biopsy. All the complications were ≤grade 3 according to the Clavien–Dindo classification system for surgical complications [18]. No patient died or was readmitted due to prostate biopsy.

**Table 3.** Prostate-biopsy-related complications according to the Clavien–Dindo classification.

| Type of Complication (Number, %) | Any Grade | Grade 3 |
| --- | --- | --- |
| Prostatitis | 2 (1.9) | 1 (1.0) |
| Hematuria | 44 (42.3) | 2 (1.9) |
| Urinary retention | 3 (2.9) | 2 (1.9) |
| Hematospermia | 1 (1.0) | - |

### 4. Discussion

According to the recent PI-RADS v2 guidelines, standard examination protocols require MRI in T2WI and DWI or DCE-MRI after injection of an intravenous contrast agent [14]. Recent studies focused on the utility of bpMRI in the diagnosis of PCa. Tan

et al. investigated and compared the accuracies of DCE-MRI and combined DWI and T2WI in diagnosing csPCa [7]. Although DCE-MRI has a higher specificity, the area under the receiver operating characteristic curve and the overall sensitivity of DCE-MRI are not significantly different from those of T2WI [7]. However, DCE-MRI was found to be less sensitive than DWI alone and combined DWI and T2WI. It was also found to be less specific than DWI alone. Furthermore, it was found that the specificity of DCE-MRI was not significantly different from that of combined DWI and T2WI [7]. Other recent studies reported no statistically significant differences in the csPCa detection rate between bpMRI and mpMRI [11,15]. Junker et al. compared bpMRI and mpMRI in 236 patients who underwent mpMRI because PCa was suspected [19]. When DCE-MRI was omitted, 94.1% of patients with PCa were found to have the same PI-RADS scores, and 5.9% of them had their PI-RADS scores downgraded from 4 to 3 [19]. Similarly, Kim et al. investigated 730 consecutive patients who underwent mpMRI before radical prostatectomy [20]. A total of 196 patients who had negative mpMRI results with no suspicious lesions in the prostate were postoperatively diagnosed with PCa [20]. The final pathological examination of the 196 patients with negative mpMRI results revealed that 6.6% of them had T3 PCa and 1.0% had lymph node involvement [20]. In addition, 1.0% of these patients had a GS of 4 + 4, 21.45% had a GS of 4 + 3, and 59.7% had a GS of 3 + 4 [20]. The table time for traditional mpMRI was approximately 45 minutes, but high diagnostic accuracy was achieved with a rapid bpMRI protocol in approximately a third of the table time without the use of a contrast agent [21]. There are also economic benefits to abandoning the standard practice of using gadolinium in prostate MRI [20]. Radtke et al. reported that bpMRI is not inferior to mpMRI in terms of csPCa detection and that bpMRI has the advantage of superior cost effectiveness [22]. Therefore, current recommendations suggest that gadolinium use should be limited to settings where it is necessary for diagnosis [23]. We suppose that omitting DCE-MRI does not lead to significant differences in diagnostic accuracy or PCa detection rates, and it appears that a biparametric approach is used for the initial routine prostate MRI with PI-RADS v2.

It is yet to be determined whether TBx or SBx alone, instead of combined TBx and SBx, is sufficient for the diagnostic evaluation of biopsy-naïve patients with suspicious lesions on MRI. Hansen et al. reported that combined TBx and SBx is significantly better than TBx or SBx alone for PCa detection in patients with GS of 7–10 who have PI-RADS v2 scores of 4 and 5 ($p < 0.001$) [24]. With regard to PI-RADS 3 lesions, the csPCa detection rate of TBx alone was found to be significantly lower than that of combined TBx and SBx ($p < 0.001$), and there were no significant differences in detection rate between SBx alone and combined TBx and SBx ($p = 0.063$) [24]. Mannaerts et al. showed that compared to TBx alone, combined TBx and SBx significantly improved PCa detection rates from 5% to 15% [25]. A diagnosis of unilateral disease was made using mpMRI in 22% of patients found to be positive for csPCa on combined TBx and SBx, while SBx detected bilateral csPCa [25]. TBx has a high sensitivity for index lesion characterization, but secondary lesions are often missed by imaging [25]. TBx did not remarkably change the PCa detection rate ($p > 0.9$) or the csPCa detection rate ($p = 0.67$) even though TBx detected PCa in 27 patients (51%), which included 22 patients (82%) with csPCa, and SBx detected csPCa in 36% of patients and clinically insignificant PCa in 15% of patients [26]. Combined TBx and SBx reduced the risk of GS increase on final histopathology by 22% [27]. Interestingly, 44.3% of patients with low-risk PCa according to TBx were reclassified as intermediate-risk PCa based on SBx results [27]. In this study, it was appropriate to detect csPCa using PI-RADS v2 based on bpMRI because of the relatively high PPV of the scoring system. However, the accuracy of csPCa diagnosis of combined TBx and SBx was significantly higher than that of TBx alone. Combined TBx and SBx may also be necessary for the detection of csPCa.

Several studies have reported complication rates following prostate biopsy [5,28]. Hematuria is reported in 10–84% of biopsies, rectal bleeding in 1–45%, infections in up to 6.3%, urinary retention in up to 1.7%, and hospitalization in up to 6.9% [5]. Of these complications, severe infection remains the most lethal, and it includes meningitis, vertebral

osteomyelitis, sepsis, and septic shock [28]. In recent years, fluoroquinolone resistance has increased globally [29], and the presence of fluoroquinolone-resistant organisms on rectal swab culture is a significant predictor of infection after prostate biopsy [30]. Infection is an increasingly important complication after biopsy. Therefore, transperineal biopsy is suggested as a possible alternative procedure to avoid infection [28]. In the future, improved markers and imaging may reduce the need for invasive biopsy procedures for many patients [5]. In this study, the complication rate after prostate biopsy was relatively low. PI-RADS v2 based on bpMRI may be an important factor to consider when deciding whether prostate biopsy should be performed for csPCa diagnosis in patients with suspicious lesions.

There are several limitations to our study. First, this was a retrospective study; therefore, it has an inherent potential for bias. Second, there was no control group of patients who received a full mpMRI protocol, and this was a nonrandomized study. Thus, it is necessary to carefully compare our results with those of previous studies. Third, not all the patients with PI-RADS v2 scores of 1 or 2 underwent prostate biopsy. Therefore, these patients may be underrepresented, and the csPCa detection rate was relatively high in this study. Finally, this study focused on the correlation of bpMRI with biopsy pathology only and not with prostatectomy specimens.

## 5. Conclusions

The diagnostic accuracy of bpMRI, including T2WI and DWI, is comparable to that of standard mpMRI for the detection of csPCa. In addition, bpMRI may reduce intravenous contrast agent-related risks, reduce examination time, and reduce costs, without significantly lowering diagnostic accuracy. Moreover, combined TBx and SBx may be optimal for the accurate determination of csPCa diagnosis, ISUP grade, risk classification, and decision-making for the treatment of PCa, particularly in daily medical practice.

**Author Contributions:** Conceptualization, D.K. and T.K.; methodology, D.K. and T.K.; investigation, D.K.; resources, K.O., S.T., M.K., K.K., C.N., M.T., K.I., K.N., H.K., and N.S.; data curation, D.K.; writing—original draft preparation, D.K.; writing—review and editing, T.K.; supervision, M.M. and T.M. All authors have read and agreed to the published version of the manuscript.

**Funding:** This research received no external funding.

**Institutional Review Board Statement:** The study protocol was approved by the Institutional Review Board of Gifu University (number: 30-031). All procedures performed in studies involving human participants were in accordance with the ethical standards of the institutional and/or national research committee and with the 1964 Helsinki declaration and its later amendments or comparable ethical standards.

**Informed Consent Statement:** For this type of study, formal consent is not required. Pursuant to the provisions of the ethics committee and the ethic guideline in Japan, written consent was not required in exchange for public disclosure of study information in the case of retrospective and/or observational study using a material such as the existing documentation. The study information was open for the public consumption at http://www.med.gifu-u.ac.jp/file/2020-064.pdf (accessed on 1 February 2021).

**Data Availability Statement:** The data presented in this study are available on request from the corresponding author. The data are not publicly available due to privacy and ethical reasons.

**Conflicts of Interest:** The authors declare no conflict of interest.

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
