# Peer review of "The Utility of Combined Target and Systematic Prostate Biopsies in the Diagnosis of Clinically Significant Prostate Cancer Using Prostate Imaging Reporting and Data System Version 2 Based on Biparametric Magnetic Resonance Imaging"

_curroncol, doi:10.3390/curroncol28020123_

Round 1

Reviewer 1 Report

Kato et al. examined the value of bpMRI for detection os csPCa in comparison with mpMRI. This is a well-written study and its limitations are appropriately discussed by the authors. Some minor comments need to be addressed, as follows:

  1. The authors seem to overstate the inconsistency between TBx and SBx in favor of the latter. It should be highlighted that while 41.5% was upgraded on SBx, a similar proportion of lesions/pts was upgraded on TBx (34.1%).
  2. In general, during daily practice it is much more likely for combined TBx and SBx to be used in a patient, either concurrently or sequentially to inform treatment decisions rather than either one (TBx or Sbx) alone.

Author Response

12, March, 2021

Dr. Amber Li

Assigned Editor

Current Oncology

Dear Editor:

Thank you very much for the review of our manuscript titled “The utility of combined target and systematic prostate biopsies in the diagnosis of clinically significant prostate cancer using Prostate Imaging Reporting and Data System version 2 based on biparametric magnetic resonance imaging.”

We sincerely appreciate all valuable comments and suggestions, which helped us to improve the quality of our manuscript. Our responses to the Reviewers’ comments are described below in a point-to-point manner. Appropriate changes, suggested by the Reviewers, have been introduced to the manuscript (track-changes mode in the red color font). Let me emphasize our full readiness to make any further improvements to the manuscript.

We hope that our manuscript will be acceptable for publication in the Cancer Reports.

We look forward to hearing from you.

Yours sincerely,

Takuya Koie

Department of Urology

Gifu University Graduate School of Medicine

1-1 Yanagido, Gifu, Gifu 501-1194, Japan

TEL.: +81-582-30-6338

FAX: +81-582-30-6341

e-mail: goodwin@gifu-u.ac.jp

Responses to the reviewer's comments

We would like to thank the Reviewers for taking the time and effort necessary to review the manuscript. We sincerely appreciate all the valuable comments and suggestions, which helped us to improve the quality of the manuscript.

Response to Reviewer 1

The authors appreciate the reviewer’s comments. The authors’ point-by-point responses to the comments are given below.

  1. The authors seem to overstate the inconsistency between TBx and SBx in favor of the latter. It should be highlighted that while 41.5% was upgraded on SBx, a similar proportion of lesions/pts was upgraded on TBx (34.1%).

Response:

We have revised the following sentence on line 30 and 125:

In addition, the rate of increase in the Gleason score (GS) was similar between SBx (41.5%) and TBx (34.1%).

  1. In general, during daily practice it is much more likely for combined TBx and SBx to be used in a patient, either concurrently or sequentially to inform treatment decisions rather than either one (TBx or Sbx) alone.

Response:

We have revised the following sentence on line 198:

risk classification and decision-making for the treatment of PCa, particularly in daily medical practice.

Reviewer 2 Report

This paper describes the detection rates of targeted and systematic prostate biopsies in patients with elevated PSA levels. 

Overall, this is a well written paper in which the conclusion appear sound. 

First, the number of patients assessed is relatively small

Some issues though remain.

1) How can the authors compare the detections rates reported in their study and using bpMRI to those of authors. The prognostic parameters may differ (PSA level, cT stage, age) and this is apparently not a randomised clinical trial

  • The detection rates of targeted and systematic prostate biopsies need to be separated as well as reported together. Which part of csPa was detected using targeted and systematic biopsies alone, and which part was detected on both biopsy techniques?
  • It is Gleason score 6-10 and ISUP grade 1-5. The authors need to make a clear distinction in this. Gleason score pattern does not exist. So csPCa needs to be Gleason score 3 + 4 = 7 and higher and/or ISUP 2 and higher.
  • Was the number of biopsy cores assessed and reported as a parameter of csPa? 
  • Were bpMRIs assessed by two independent radiologists?
  • Could the authors clarify how the targeted biopsies were taken as either cognitive, MRI-directed, otherwise?

Author Response

12, March, 2021

Dr. Amber Li

Assigned Editor

Current Oncology

Dear Editor:

Thank you very much for the review of our manuscript titled “The utility of combined target and systematic prostate biopsies in the diagnosis of clinically significant prostate cancer using Prostate Imaging Reporting and Data System version 2 based on biparametric magnetic resonance imaging.”

We sincerely appreciate all valuable comments and suggestions, which helped us to improve the quality of our manuscript. Our responses to the Reviewers’ comments are described below in a point-to-point manner. Appropriate changes, suggested by the Reviewers, have been introduced to the manuscript (track-changes mode in the red color font). Let me emphasize our full readiness to make any further improvements to the manuscript.

We hope that our manuscript will be acceptable for publication in the Cancer Reports.

We look forward to hearing from you.

Yours sincerely,

Takuya Koie

Department of Urology

Gifu University Graduate School of Medicine

1-1 Yanagido, Gifu, Gifu 501-1194, Japan

TEL.: +81-582-30-6338

FAX: +81-582-30-6341

e-mail: goodwin@gifu-u.ac.jp

Responses to the reviewer's comments

We would like to thank the Reviewers for taking the time and effort necessary to review the manuscript. We sincerely appreciate all the valuable comments and suggestions, which helped us to improve the quality of the manuscript.

Response to Reviewer 2

The authors appreciate the reviewer’s comments. The authors’ point-by-point responses to the comments are given below.

  1. How can the authors compare the detections rates reported in their study and using bpMRI to those of authors. The prognostic parameters may differ (PSA level, cT stage, age) and this is apparently not a randomised clinical trial

Response:

We completely agree with your opinion.

Therefore, we have added the following sentence on line 190.

Thus, it is necessary to carefully compare our results with those of previous studies.

  1. The detection rates of targeted and systematic prostate biopsies need to be separated as well as reported together. Which part of csPa was detected using targeted and systematic biopsies alone, and which part was detected on both biopsy techniques

Response:

We have added the following words on 114:

combined TBx and SBx was significantly superior to TBx (69.2%) or SBx (63.5%) alone.

  1. It is Gleason score 6-10 and ISUP grade 1-5. The authors need to make a clear distinction in this. Gleason score pattern does not exist. So csPCa needs to be Gleason score 3 + 4 = 7 and higher and/or ISUP 2 and higher.

Response:

We have revised the following sentence on line 92:

csPCa was defined by the International Society of Urological Pathology (ISUP) grade ≥2

We have revised the following sentence on line 116:

According to the PCa detection, the association between PI-RADS v2 and ISUP grade is shown in Fig. 1. PI-RADS v2 scores were found to significantly correlate with ISUP grade (p = 0.004; Fig. 1).

We have revised the following sentence on line 121:

The association between Prostate Imaging Reporting and Data System version 2 (PI-RADS v2) and ISUP grade according to the prostate cancer detection. PI-RADS v2 scores were found to significantly correlate with ISUP grade (p = 0.004).

We have revised the following sentence on line 128:

TABLE 2. ISUP grade and prostate cancer location concordance outcomes according to target or systemic biopsies in patients with clinical significant prostate cancer

We have also revised Table 2:

Concordance ISUP grade between target and systemic biopsy

Upgrading ISUP grade on target biopsy

Upgrading ISUP grade on systemic biopsy

  1. Was the number of biopsy cores assessed and reported as a parameter of csPa?

Response:

We have revised the following sentence on line 92:

and/or a maximum cancer core length ≥4 mm within at least one specimen, which was obtained after MRI-TRUS fusion biopsies, whereas clinically

insignificant cancer was defined as group 1 according to the 2014 ISUP guidelines [6,17].

  1. Were bpMRIs assessed by two independent radiologists?

Response:

We have added the following sentence on line 78.

At least two experienced radiologists (with more than 5 years of experience in PCa diagnosis) were assigned to review the bpMRI.

  1. Could the authors clarify how the targeted biopsies were taken as either cognitive, MRI-directed, otherwise?

Response:

We have revised the following sentence on line 86:

Each TBx was performed using an MRI-TRUS fusion biopsy.

Round 2

Reviewer 2 Report

The authors responded sufficiently to the comments of the reviewer